# Course and Lethality of SARS-CoV-2 Epidemic in Nursing Homes after Vaccination in Florence, Italy

**DOI:** 10.3390/vaccines9101174

**Published:** 2021-10-13

**Authors:** Giulia Rivasi, Matteo Bulgaresi, Enrico Mossello, Primo Buscemi, Chiara Lorini, Daniela Balzi, Riccardo Barucci, Ilaria Del Lungo, Salvatore Gangemi, Sante Giardini, Cecilia Piga, Eleonora Barghini, Serena Boni, Giulia Bulli, Paolo Carrai, Andrea Crociani, Antonio Faraone, Aldo Lo Forte, Letizia Martella, Simone Pupo, Giacomo Fortini, Irene Marozzi, Giulia Bandini, Claudia Cosma, Lorenzo Stacchini, Gabriele Vaccaro, Lorenzo Baggiani, Giancarlo Landini, Guglielmo Bonaccorsi, Andrea Ungar, Enrico Benvenuti

**Affiliations:** 1Division of Geriatric and Intensive Care Medicine, Careggi Hospital and University of Florence, 50134 Florence, Tuscany, Italy; enrico.mossello@unifi.it (E.M.); irene.marozzi@libero.it (I.M.); aungar@unifi.it (A.U.); 2Geriatric Unit, Santa Maria Annunziata Hospital, Local Health Unit “Toscana Centro”, 50134 Florence, Tuscany, Italy; matteo.bulgaresi@uslcentro.toscana.it (M.B.); riccardo.barucci@uslcentro.toscana.it (R.B.); ilaria.dellungo@uslcentro.toscana.it (I.D.L.); salvatore.gangemi@uslcentro.toscana.it (S.G.); sante.giardini@uslcentro.toscana.it (S.G.); cecilia.piga@uslcentro.toscana.it (C.P.); eleonora.barghini@uslcentro.toscana.it (E.B.); serena.boni@uslcentro.toscana.it (S.B.); giulia2.bulli@uslcentro.toscana.it (G.B.); letizia.martella@uslcentro.toscana.it (L.M.); simone.pupo@uslcentro.toscana.it (S.P.); giacomo.fortini@uslcentro.toscana.it (G.F.); enrico.benvenuti@uslcentro.toscana.it (E.B.); 3Department of Health Science, University of Florence, 50134 Florence, Tuscany, Italy; primo.buscemi@unifi.it (P.B.); chiara.lorini@unifi.it (C.L.); claudia.cosma@unifi.it (C.C.); lorenzo.stacchini@unifi.it (L.S.); gabriele.vaccaro@unifi.it (G.V.); guglielmo.bonaccorsi@unifi.it (G.B.); 4Epidemiology Unit, Local Health Unit “Toscana Centro”, 50134 Florence, Tuscany, Italy; daniela.balzi@uslcentro.toscana.it; 5Department of Internal Medicine, San Giovanni di Dio Hospital, 50134 Florence, Tuscany, Italy; paolo.carrai@uslcentro.toscana.it (P.C.); andrea.crociani@uslcentro.toscana.it (A.C.); antonio.faraone@uslcentro.toscana.it (A.F.); aldo.loforte@uslcentro.toscana.it (A.L.F.); 6Division of Internal Medicine, Careggi Hospital, 50134 Florence, Tuscany, Italy; giulia.bandini@unifi.it; 7Department of Territorial Health Network, 50134 Florence, Tuscany, Italy; lorenzo.baggiani@uslcentro.toscana.it; 8Department of Internal Medicine, Santa Maria Nuova Hospital, Local Health Unit “Toscana Centro”, 50134 Florence, Tuscany, Italy; giancarlo.landini@uslcentro.toscana.it

**Keywords:** COVID-19, vaccine, mortality, hospitalization, lethality, nursing home residents, older adults

## Abstract

Evidence on the effectiveness of SARS-CoV-2 vaccines in nursing home (NHs) residents is limited. We examined the impact of the BNT162b2 mRNA SARS-CoV-2 vaccine on the course of the epidemic in NHs in the Florence Health District, Italy, before and after vaccination. Moreover, we assessed survival and hospitalization by vaccination status in SARS-CoV-2-positive cases occurring during the post-vaccination period. We calculated the weekly infection rates during the pre-vaccination (1 October–26 December 2020) and post-vaccination period (27 December 2020–31 March 2021). Cox analysis was used to analyze survival by vaccination status. The study involved 3730 residents (mean age 84, 69% female). Weekly infection rates fluctuated during the pre-vaccination period (1.8%–6.5%) and dropped to zero during the post-vaccination period. Nine unvaccinated (UN), 56 partially vaccinated (PV) and 35 fully vaccinated (FV) residents tested SARS-CoV-2+ during the post-vaccination period. FV showed significantly lower hospitalization and mortality rates than PV and UV (hospitalization: FV 3%, PV 14%, UV 33%; mortality: FV 6%, PV 18%, UV 56%). The death risk was 84% and 96% lower in PV (HR 0.157, 95%CI 0.049–0.491) and FV (HR 0.037, 95%CI 0.006–0.223) versus UV. SARS-CoV-2 vaccination was followed by a marked decline in infection rates and was associated with lower morbidity and mortality among infected NH residents.

## 1. Introduction

The COVID-19 pandemic has dramatically affected nursing home (NH) residents [1,2,3,4], thus calling for the early vaccination of this vulnerable population.

It is known that the immune response is reduced at an advanced age even in healthy older people, due to age-related changes in immune functions which are commonly referred to as immunosenescence [5]. As a consequence, vaccinations may trigger less effective immunity in older adults [6]. In the NH setting, multimorbidity, medications and poor nutritional status may contribute to further impairing immune responses, thereby hampering vaccine effectiveness. Indeed, previous studies indicated poor immunogenicity of influenza vaccines in NH residents [7,8].

The results of clinical trials on SARS-CoV-2 vaccines have reported comparable efficacy in older and younger adults [9,10,11]. However, it should be considered that the oldest patients (e.g., above the age of 75) were poorly represented in the study populations, and the exclusion criteria included a wide range of conditions which are very common at advance ages, such as frailty, dementia and multimorbidity [12,13]. The immunogenicity of the BNT162b2 mRNA vaccine was found to be lower in adults aged 65–85 years than in younger people [14]. Moreover, it is unclear if and to what extent vaccination, beyond decreasing COVID-19 clinical expression, would be effective in limiting the spread of SARS-CoV-2 infection in closed communities of vulnerable subjects, such as NHs [13]. Limited post-authorization data are available on SARS-CoV-2 vaccine efficacy and effectiveness in NH residents [12], and some preliminary observations suggest a blunted antibody response in this population [15,16]. Given these premises, the effects of SARS-CoV-2 vaccine in the NH setting remain largely unexplored.

The primary aim of this study was to examine the impact of the BNT162b2 mRNA SARS-CoV-2 vaccine on the course of the epidemic in the NHs of the Florence Health District in Tuscany, Italy, through the analysis of weekly infection rates before and after the vaccine campaign. The secondary aim was to assess death and hospitalization risks by vaccination status in residents testing SARS-CoV-2+ in NHs where at least one vaccine dose had been administered to most residents.

## 2. Materials and Methods

On 21 December 2020, the European Medicines Agency (EMA) granted emergency authorization for the BNT162b2 (Pfizer-BioNTech, New York, NY, USA) SARS-CoV-2 mRNA vaccine. Together with healthcare workers, SARS-CoV-2-naïve NH residents were the first category to receive the vaccine in Florence Health District, Tuscany, Italy. The two doses of BNT162b2 were administered 21 days apart during the periods 27 December 2020–17 January 2021 (first dose) and 21 January–15 February 2021 (second dose). Consent to vaccination was provided by residents or their legal representatives. In cases where residents were unable to provide their consent and legal representatives were unavailable, vaccination was performed with prior medical authorization, according to the provisions of Legislative Decree 1/2021 (5 January 2021). During the same period, the BNT162b2 mRNA vaccine was administered to all staff members, who provided their consent.

At the beginning of the COVID-19 pandemic, an innovative healthcare and organizational model was developed in Tuscany, involving a hospital-at-nursing-home multidisciplinary team (GIROT, Gruppo Intervento Rapido Ospedale Territorio) aiming to provide on-site intermediate care assistance to NH residents affected by COVID-19.

A detailed description of the GIROT model has been outlined elsewhere [17]. In brief, intermediate care units were created on-site in the NHs with active COVID-19 outbreaks and infected residents’ care was provided by GIROT medical specialists and nurses, in collaboration with general practitioners and non-specialist medical doctors from Continuity Care Teams. The tasks of GIROT included the following: infection transmission control among NH residents and staff; comprehensive geriatric assessments including risk stratification and prognostication to define treatment goals and avoid unnecessary hospital admissions; on-site diagnostic assessments and protocol-based treatment of COVID-19; the management of geriatric syndromes, such as delirium, constipation, malnutrition, and pressure sores; the supply of nursing personnel to understaffed NHs. In the Florence Health District, the activity of GIROT started during the first wave of the pandemic (March–April 2020) and was then optimized during the second wave (October 2020–January 2021) with the standardization of operative protocols.

In this context, since 1 October 2020, all NH residents and healthcare workers in the Florence Health District have undergone SARS-CoV-2 infection screening with antigenic swabs every 2 weeks. If at least one positive case was detected, all residents and workers of the index NH received a polymerase chain reaction test with the aim of identifying incident infections and limiting transmission, isolating SARS-CoV-2-positive residents within dedicated areas (“COVID-19 bubbles”). During the study period, infection management was standardized based on the GIROT protocols. Transmission control strategies including the use of personal protective equipment and hygiene measures were also maintained after the vaccine campaign had been completed. Visitors were not allowed to enter the NHs during the study period.

For the purpose of this study, we conducted a retrospective analysis of GIROT data from the NHs of the Florence Health District involved in the second wave of the pandemic. We estimated the weekly SARS-CoV-2 infection rates from 1 October 2020 to 26 December 2021 (the pre-vax period) and from 27 December to 31 March 2021 (the post-vax period, after the beginning of the vaccination campaign). The 7-day incidence of new cases of SARS-CoV-2 infections was calculated as the number of new confirmed SARS-CoV-2-positive residents over the SARS-CoV-2-naïve resident census. SARS-CoV-2 cases were diagnosed based on SARS-CoV-2 nucleic acid amplification tests from a nasopharyngeal swab.

Moreover, we recorded the main clinical features of new SARS-CoV-2+ cases occurring in NHs where at least one dose of the vaccine had been administered to most residents. These subjects were divided into three groups according to vaccination status at the time of positive swab detection: (1) unvaccinated (had never received a COVID-19 vaccine dose); (2) partially vaccinated (had received only the first dose); and (3) fully vaccinated (had received two doses of a two-dose series). For each subject, their age and gender, and the presence and severity of COVID-19 symptoms were reported. The latter were categorized as: absent/minimal, mild (fever, cough, flu-like symptoms, diarrhea) or severe (respiratory failure requiring oxygen therapy, severe gastrointestinal symptoms including severe anorexia, delirium). Hospital admissions and deaths (due to COVID-19 or other causes) were also recorded.

### Statistical Analysis

Patient characteristics were summarized using medians and interquartile ranges (IQRs) for continuous variables and percentages for categorical variables. The two-tailed Fisher exact test was used to test differences in residents’ characteristics and outcomes (sex, age, hospitalization, symptoms, GIROT severity scale, death) according to vaccination status (unvaccinated, partially vaccinated and fully vaccinated). Kaplan–Meier survival curves were generated based on vaccination status and the log-rank test was used to assess differences between curves. The risk of COVID-19 related death was assessed using a Cox proportional hazards regression model including age, sex and vaccination status, with unvaccinated residents as the reference category. Associations are presented as hazard ratios (HR) with 95% confidence intervals (CI). All statistical analyses were performed using R (Version 4.0.1).

## 3. Results

Among 3730 SARS-CoV-2-naïve residents living in Florence Health District NHs as of 1 October 2020 (mean age 84.2 years, 69% female), a total of 1658 cases of SARS-CoV-2 infections were confirmed during the whole study period.

In the pre-vax period, weekly infection rates fluctuated between a nadir of 1.8% and a peak of 6.5%. In the post-vax period, infection rates progressively decreased from 4.5% to zero, then remained stable until the end of the observation period. This trend is illustrated in Figure 1, in comparison with new SARS-CoV-2 cases reported in the general population of Tuscany region during the same observation period.

During the post-vax period, 100 new SARS-CoV-2 cases were recorded in six NHs, where at least one vaccine dose had been administered, including nine unvaccinated, 56 partially vaccinated and 35 fully vaccinated residents. Their median age was 86 years and 67% were female. Demographic and clinical characteristics by vaccination status are detailed in Table 1.

Unvaccinated subjects were significantly younger than vaccinated ones (*p* < 0.001). Fully vaccinated residents were mainly asymptomatic (86%) or reported mild symptoms. Partially vaccinated and unvaccinated residents reported COVID-19 symptoms in 70% and 78% of cases, respectively, with 44% of unvaccinated subjects developing severe disease. The presence and severity of symptoms showed a significant difference (*p* < 0.001) across vaccination groups. Hospitalization for COVID-19 was significantly more common among unvaccinated residents (33%) as compared to partially and fully vaccinated ones (14% and 3%, respectively, *p* = 0.04). A total of 17 COVID-19-related deaths occurred. The median time interval from infection diagnosis to death was 16 days (IQR = 7–23 days). The COVID-19 mortality rate was significantly higher among unvaccinated subjects (56%) as compared to partially (18%) and fully vaccinated ones (6%) (*p* < 0.001).

The survival curves of SARS-CoV-2+ residents differed significantly according to vaccination status (*p* < 0.001 at the log-rank test, Figure 2), with unvaccinated subjects showing a higher risk of COVID-19-related mortality compared to partially and fully vaccinated residents (*p* = 0.003 and *p* < 0.001 in the post hoc comparison). The mortality risk was similar in partially and fully vaccinated SARS-CoV-2+ residents (*p* > 0.05).

In a Cox proportional hazards regression model (Table 2), vaccination status was significantly and independently associated with COVID-19 mortality, with the death risk being 84% and 96% lower in partially and fully vaccinated residents, respectively, vs. unvaccinated residents (HR 0.157, 95%CI 0.049–0.491, *p* = 0.002 in partially vaccinated and HR 0.037, 95%CI 0.006–0.223, *p* < 0.001 in fully vaccinated residents). Age was significantly associated with COVID-19-related mortality only when adjusting for sex and vaccination status (HR 1.09, 95%CI 1.02–1.17).

## 4. Discussion

The results of pivotal clinical trials have demonstrated near 95% effectiveness of the BNT162b2 mRNA vaccine in preventing symptomatic COVID-19, with new SARS-CoV-2 infections starting to decrease as early as two weeks after the administration of the first dose (52% effectiveness), corresponding to the time at which the antibody titer increases to protective levels [9]. The available data referring to NH residents suggest a blunted immune response to the BNT162b2 mRNA vaccine, with nearly 4-fold lower median neutralization titers than those observed in healthcare workers [15]. A combined humoral and cellular immune response can be detected in 37% of SARS-CoV-2-naïve residents as compared to 87% of SARS-CoV-2-naïve healthcare workers four weeks after the first dose [16]. Moreover, approximately 17% of residents show neutralizing titers at or below the lower limit of detection two weeks after the second dose [15]. These data raise the possibility that the SARS-CoV-2 mRNA vaccine might have reduced effectiveness in protecting NH residents from the infection.

In our study sample, we observed a sharp decrease in SARS-CoV-2 weekly infection rates over the three-month post-vax period. By contrast, the number of new SARS-CoV-2 cases substantially increased in the general population of this county during the observation period, thus constituting the third wave of the pandemic. Similar data have been reported in a sample of NHs in Spain and in a large sample of US NHs, in which an impressive decline in SARS-CoV-2 new cases was observed after vaccination, running counter to the pattern of the local general population [18,19]. We may thus infer that the BNT162b2 mRNA vaccine significantly contributed to reducing the rate of new SARS-CoV-2 cases among NH residents and protected this highly vulnerable population from the third wave of the pandemic. As further confirmation, a recent report showed an accelerated decline of incident infections in early vaccinated US NHs as compared to facilities that were waiting for vaccination [20]. In particular, in the first and second week after the vaccination, early vaccinated NHs showed an absolute 1.6% and 2.5% decrease in incident infections, respectively, in comparison with the expected infections. Moreover, a cumulative incidence reduction of 5.2% was estimated during the following 5 weeks [20].

In our study sample, the effects of vaccination were already detectable before the start of the administration of the second dose and became evident over the follow-up time, with the infection rate falling to zero 8 weeks after the beginning of the vaccination campaign (Figure 1). A similar trend has been described in a report on 130 Veterans Affairs Community Living Centers, in which the proportion of SARS-CoV-2+ tests dropped four weeks after the first vaccine dose and continued afterward, and this was also observed among unvaccinated residents [21]. Consistently with these findings, data from 2501 US NHs indicate that the spread of SARS-CoV-2 significantly decreased five and six weeks after the first BNT162b2 vaccine dose, both among residents and staff [22]. Notably, among asymptomatic NH residents testing SARS-CoV-2+ at surveillance controls, single-dose mRNA vaccination was found to be associated with a lower nasopharyngeal viral load than that detected in unvaccinated subjects [23]. This finding suggests that the vaccine may have immediate benefit on infection transmission, even before full immunization is reached, which could be of great clinical relevance in this congregate setting.

According to the administrative data of the Florence Health District, vaccination coverage reached 97% of residents and 85% of staff members in the NHs included in the present study. Although previous studies have suggested reduced immunogenicity of SARS-CoV-2 vaccine in NH residents, high vaccination coverage of residents and staff has probably allowed for the protection of those residents who might have a suboptimal immune response to the vaccine. Indeed, staff vaccination has been shown to reduce the spread of SARS-CoV-2 infection [22], contributing to the preserving of subjects who had not achieved protective immunity after the vaccine. NH staff vaccination thus represents an example of “cocooning vaccination”, as recently described for COVID-19 in cancer patients [24]. Finally, it has to be stressed that the retention of transmission control strategies and visiting restrictions can partially explain the prevention of the third wave of the epidemic in the NH setting.

In addition to positively impacting the spread of the infection, in our study sample SARS-CoV-2 vaccination was also associated with reduced COVID-19 symptoms, hospital admissions and mortality, in spite of the older age of the vaccinated subjects. Indeed, among SARS-CoV-2+ fully-vaccinated residents (n = 35), the majority was asymptomatic (86%) or reported mild symptoms, only one was hospitalized (3%) and two died (6%). Of them, six subjects were fully vaccinated but not immune at the time of the SARS-CoV-2 infection, as <14 days had elapsed since the second dose had been administered [25,26]. Similar results were reported in 15 skilled nursing facilities in Chicago, where a routine screening identified 12 cases of SARS-CoV-2 infection among fully-vaccinated residents (i.e., ≥14 days after the second dose): most of them were asymptomatic (8/12), two were hospitalized and one died [25]. Conversely, among the small number of unvaccinated residents (n = 9), 44% developed severe symptoms, hospital admission was necessary in three cases (33%) and five (56%) died (Table 1).

Although COVID-19-related mortality was considerable among the SARS-CoV-2+ vaccinated residents of our sample (12/91, 13%), vaccination was found to significantly improve survival, even after one single dose, with an 84% reduction in the mortality risk and a 96% reduction in partially and fully vaccinated residents, respectively. Moreover, COVID-19-related mortality among vaccinated residents was substantially lower as compared to that observed in the same population before the vaccine campaign (23%) [17]. Our results are consistent with the recent reports from the US NHs, showing that vaccination is associated with lower COVID-19 mortality rates among residents, starting from four weeks after the vaccine [20,22].

The findings of this study are subject to some limitations. First, the limited observation period did not allow us to analyze the long-term impact of vaccination on the course of the epidemic. Moreover, the antibody response to the vaccine was not assessed, thus preventing us from correlating infection rates and COVID-19 severity with serum antibody titers. Finally, variant strains were not systematically investigated in our study sample. However, some specimens were submitted for genotyping and cases caused by SARS-CoV-2 variants were detected during the post-vax period in the NHs involved in the study, as well as in the general population of the Florence Health District, suggesting that variants were circulating in the area during the observation period.

## 5. Conclusions

In nursing home residents of the Florence Health District, SARS-CoV-2 vaccination was followed by a marked decline of infection rates and was associated with lower morbidity and mortality among infected residents, thus confirming the effectiveness of the SARS-CoV-2 vaccine in this vulnerable population. The present data lend support to the necessity of a high SARS-CoV-2 vaccination coverage among NH residents.

## Figures and Tables

**Figure 1 vaccines-09-01174-f001:**
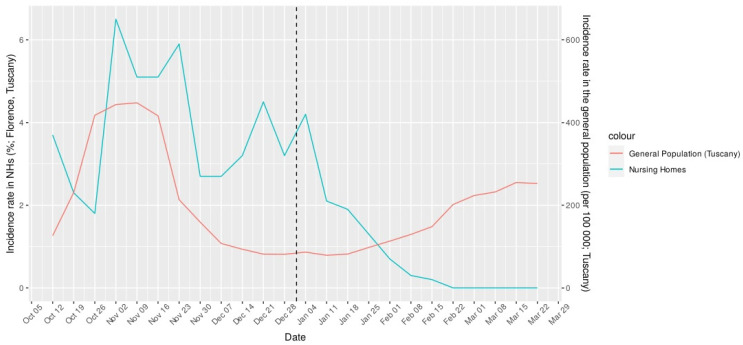
The weekly incidence rate of COVID-19 in NHs of the Florence Health District (per 100 people; blue line) and in the general population of Tuscany (per 100,000 people; red line), before and after the start of the vaccination campaign (dashed vertical line). Data on the general population were obtained from the Italian Ministry of Health. The date on the *x*-axis indicates the first day of the monitoring week.

**Figure 2 vaccines-09-01174-f002:**
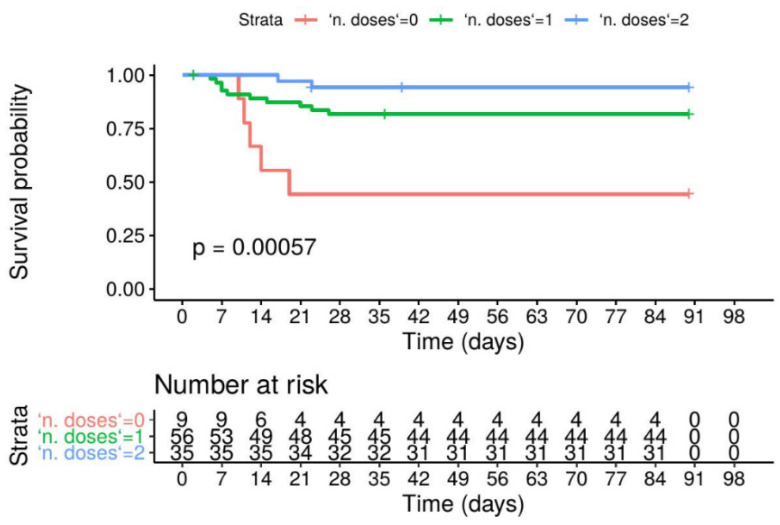
Kaplan–Meier survival estimates for COVID-19-related mortality depending on vaccination status (unvaccinated, partially vaccinated and fully vaccinated residents, log-rank test).

**Table 1 vaccines-09-01174-t001:** Demographic and clinical characteristics of SARS-CoV-2 cases, according to vaccination status.

	All	Unvaccinated	Partially Vaccinated	Fully Vaccinated	*p*
	(N = 100)	(N = 9)	(N = 56)	(N = 35)	
Sex					0.170
Female	67 (67%)	5 (56%)	35 (63%)	28 (80%)	
Male	33 (33%)	4 (44%)	21 (38%)	7 (20%)	
Median age (IQR)	86.0 (79.0–91.0)	74.5 (71.7–82.5)	85.0 (80.2–90.0)	89.0 (80.0–92.5)	<0.001
Symptoms					<0.001
Yes	51 (51%)	7 (78%)	39 (70%)	5 (14%)	
No	49 (49%)	2 (22%)	17 (30%)	30 (86%)	
Symptoms severity					<0.001
Absent/minimal	57 (57%)	2 (22%)	25 (45%)	30 (86%)	
Mild	30 (30%)	3 (33%)	22 (39%)	5 (14%)	
Severe	13 (13%)	4 (44%)	9 (16%)	0 (0%)	
COVID-19 hospitalization					0.040
Yes	12 (12%)	3 (33%)	8 (14%)	1 (3%)	
No	88 (88%)	6 (67%)	48 (86%)	34 (97%)	
COVID-19 deaths					<0.001
Yes	17 (17%)	5 (56%)	10 (18%)	2 (6%)	
No	83 (83%)	4 (44%)	46 (82%)	33 (94%)	

**Table 2 vaccines-09-01174-t002:** Cox proportional hazards regression analysis of variables associated with COVID-19 related mortality in 100 SARS-CoV-2+ residents.

Variables	Unadjusted Model		Adjusted Model	
	Hazard Ratio (95%CI)	*p*	Hazard Ratio (95%CI)	*p*
Sex		0.93		0.43
Female	0.65 (0.25–1.7)		0.67 (0.25–1.81)	
Male	-		-	
Age (years)	1.0 (0.98–1.1)	0.16	1.09 (1.02–1.17)	0.013
Number of doses				
0	-		-	
1	0.26 (0.09–0.76)	0.014	0.16 (0.05–0.49)	0.002
2	0.07 (0.01–0.38)	0.002	0.04 (0.006–0.22)	<0.001

## Data Availability

Data are available upon request.

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
