# Peer review of "Course and Lethality of SARS-CoV-2 Epidemic in Nursing Homes after Vaccination in Florence, Italy"

_vaccines, 2021, doi:10.3390/vaccines9101174_

Round 1

Reviewer 1 Report

Rivasi et al. present a well-written, scientifically sound study on a very important subject: the effect of SARS-CoV2 vaccination of residents in nursing homes on infection rates, disease severity and death due to Covid-19.

Introduction: Line 57: "...it is unclear if and at to what extent..." the word "at" should be deleted.

MM: line 89: please state more clearly what you mean by "molecular swab"

Results: line 127: "Among 3730..." should read 3,730

Line 149: "symptoms in the 70%..." the word "the" should be deleted.

Figure 2: maybe use a different design in the lines e.g. dashed, dotted etc. so the figure would be easier to read even in black and white.

Figure 3 reproduces table 2 and the text. I feel it could be deleted.

Author Response

Reviewer #1

Rivasi et al. present a well-written, scientifically sound study on a very important subject: the effect of SARS-CoV2 vaccination of residents in nursing homes on infection rates, disease severity and death due to Covid-19.

Introduction: Line 57: "...it is unclear if and at to what extent..." the word "at" should be deleted.

MM: line 89: please state more clearly what you mean by "molecular swab"

Results: line 127: "Among 3730..." should read 3,730

Line 149: "symptoms in the 70%..." the word "the" should be deleted.

Figure 2: maybe use a different design in the lines e.g. dashed, dotted etc. so the figure would be easier to read even in black and white.

Figure 3 reproduces table 2 and the text. I feel it could be deleted.

We thank the Reviewer for appreciating our research and providing advice to improve our manuscript. The text has been modified as suggested. More in detail:

- “molecular swab” has been replaced by “polymerase chain reaction test” (Materials & Methods, line 89)

- Figure 2 has been modified as suggested, thus being suitable for conversion to black and white

- Figure 3 has been removed

- spell/syntax observations: modifies as suggested.

Reviewer 2 Report

This paper definitely should be added to the literature as we move through this pandemic. The elderly, rather or not in Nursing Homes, are affected the most, and it needs documentation like this article provides. The authors offered an adequate description of the demographics and clinical characteristics of the subjects from the Gruppo Intervento Rapido Ospedale Territorio (GIROT).  Adding an understanding of the location would help in the description of the subjects. An explanation to the GIROT model needed a thorough explanation rather than just referencing it from other works.  The reader is left to wonder what that model is and why it might be important in understanding this work.

The statistical analysis using the Fisher Exact test seemed appropriate for the subjects and their characteristics.  Also, the Cox proportional hazard regression provided the significance needed to believe the vaccinations succeeded in saving lives in the elderly population.  The authors gave no statistics to present the counterfactual of nursing home residents before this pandemic.  Those statistics would be helpful by describing what is the mortality rate for this population without a pandemic.

Even though the article was relatively short, it gave enough details to understand that the spread of COVID-19 can be controlled by providing a vaccination, at least in congregate settings.  The author adequately presented the findings in this paper, contributing to understanding the literature of this 21st-century pandemic.

There were just a few English editorials to make, i.e., line 52 (the oldest).  I liked the way they decided to do their statistical analysis.  This definitely should be added to the literature as we move through this pandemic.  The elderly, rather or not in NHs, are affected the most, and it needs documentation like this article provides.

Author Response

Reviewer #2

This paper definitely should be added to the literature as we move through this pandemic. The elderly, rather or not in Nursing Homes, are affected the most, and it needs documentation like this article provides. The authors offered an adequate description of the demographics and clinical characteristics of the subjects from the Gruppo Intervento Rapido Ospedale Territorio (GIROT).  Adding an understanding of the location would help in the description of the subjects. An explanation to the GIROT model needed a thorough explanation rather than just referencing it from other works.  The reader is left to wonder what that model is and why it might be important in understanding this work.

We thank the Reviewer for highlighting the epidemiological and clinical relevance of our research and the impact of the GIROT activity. We agree that the reader may appreciate additional information on the GIROT model and we have now provided a brief description in the methods section. Reference to a previous paper illustrating the model has however been maintained, as details have been omitted for brevity reasons.

In the text (Materials and Methods, from line 82): “At the beginning of COVID-19 pandemics, an innovative health care and organizational model was developed in Tuscany, involving a hospital-at-nursing home multidisciplinary team (GIROT, Gruppo Intervento Rapido Ospedale Territorio) aimed to provide on-site intermediate care assistance to NH residents affected by COVID-19. A detailed description of GIROT model has been outlined elsewhere [17].

(lines 86-98) In brief, intermediate care units were created on-site in the NHs with active COVID-19 outbreaks and infected residents’ care was provided by the GIROT medical specialists and nurses, in collaboration with general practitioners and non-specialist medical doctors from Continuity Care Teams. The GIROT tasks included the following: infection transmission control among NHs residents and staff; comprehensive geriatric assessment including risk stratification and prognostication to define treatment goals and avoid unnecessary hospital admissions; on-site diagnostic assessment and protocol-based treatment of COVID-19; management of geriatric syndromes such as delirium, constipation, malnutrition, and pressure sores; supply of nursing personnel to understaffed NHs. In Florence Health District, the GIROT activity started during the first wave of the pandemic (March-April 2020) and was then optimized during the second (October 2020 – January 2021) with the standardization of operative protocols.

(lines 99-108) In this context, since October 1st, 2020, all NH residents and healthcare workers in Florence Health District have undergone a SARS-Cov2 infection screening with antigenic swabs every 2 weeks. If at least one positive case was detected, all residents and workers of the index NH received a molecular swab polymerase chain reaction test with the aim to identify incident infections and limit transmission, isolating SARS-CoV2 positive residents within dedicated areas (“COVID-19 bubbles”). During the study period, infection management was standardized based on the GIROT protocols. Transmission control strategies including use of personal protective equipment and hygiene measures were maintained also after the vaccine campaign had been completed. Visitors were not allowed to enter the NHs all over the study period.”

The statistical analysis using the Fisher Exact test seemed appropriate for the subjects and their characteristics.  Also, the Cox proportional hazard regression provided the significance needed to believe the vaccinations succeeded in saving lives in the elderly population.  The authors gave no statistics to present the counterfactual of nursing home residents before this pandemic.  Those statistics would be helpful by describing what is the mortality rate for this population without a pandemic.

We appreciate the Reviewer’s positive feedback on the analysis. Unfortunately, we are unable to provide detailed data on mortality rates in the NHs of Florence Health District before the pandemic period. Indeed, accurate data collection started in April 2020, in parallel with the creation and development of the GIROT model. However, we considered that comparison with the mortality rates reported during the first wave of the pandemic (before the vaccine campaign), were more relevant to the purpose of our study, i.e., to understand the impact of vaccination on the epidemic course and residents’ death risk during the COVID-19 outbreak.

Even though the article was relatively short, it gave enough details to understand that the spread of COVID-19 can be controlled by providing a vaccination, at least in congregate settings.  The author adequately presented the findings in this paper, contributing to understanding the literature of this 21st-century pandemic.

There were just a few English editorials to make, i.e., line 52 (the oldest).  I liked the way they decided to do their statistical analysis.  This definitely should be added to the literature as we move through this pandemic.  The elderly, rather or not in NHs, are affected the most, and it needs documentation like this article provides.

Again, we thank the Reviewer for these positive comments on our research. Line 52 has been modified as requested.